# "We Just Want to Be Treated Normally and to Have That Healthcare That Comes along with It": Rainbow Young People's Experiences of Primary Care in Aotearoa New Zealand

Alex Ker, Tracey Gardiner, Rona Carroll *, Sally B. Rose, Sonya J. Morgan, Susan M. Garrett and Eileen M. McKinlay

Department of Primary Health Care and General Practice, University of Otago Wellington, Wellington 6242, New Zealand
* Correspondence: rona.carroll@otago.ac.nz

**Abstract:** There is growing recognition that primary care provision plays a pivotal role in improving health outcomes for LGBTQIA+ (rainbow) youth, but few studies have centered on youth experiences of primary care in Aotearoa New Zealand. This study aimed to explore the experiences and perspectives of rainbow youth when engaging with primary care. Two focus groups were held in 2021 with eleven rainbow young people aged 13–23 years recruited with assistance from local rainbow support organizations in Aotearoa NZ. Groups were audio-recorded, transcribed and analyzed using thematic analysis. Four main themes were identified: (i) anticipated and enacted discrimination, (ii) building trust, (iii) confidentiality and (iv) healthcare provider knowledge and competence. Participants spoke of some positive experiences but perceived these to be lucky or surprising, with most also having encountered less supportive interactions and heteronormative views. Visual indicators of rainbow-friendliness in clinic settings were appreciated only if affirmed by inclusive and accepting practice. Frustrations were expressed about gaps in provider knowledge and the sense of having to educate clinicians about rainbow-specific health issues. Primary care providers can facilitate positive consultations with rainbow young people by using clear communication to build trusting relationships, and by being accepting, non-judgmental and transparent about confidentiality.

**Keywords:** primary care; LGBTIA+; rainbow; focus groups; youth; healthcare experience; transgender

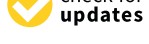

## 1. Introduction

Primary care services are the first point of contact for people seeking routine healthcare in their communities [1]. There is growing recognition that primary care provision in Aotearoa New Zealand (NZ) plays a pivotal role in improving health outcomes for rainbow youth, a diverse, often marginalized and underserved population. The term 'rainbow' is commonly used in Aotearoa NZ in place of LGBTQIA+ to broadly describe people of diverse sexualities, genders and sex characteristics. It is considered to be more inclusive of identities not captured in the acronym, particularly Māori and Pacific indigenous genders and sexualities, and culturally specific identities.

It is estimated that around one in seven secondary school-aged people (13–18 years) in Aotearoa NZ are gender- or sexuality-diverse, or are questioning if they are [2,3]. In addition to accessing primary care for general health issues, rainbow youth may also seek care for issues related to their gender or sexuality, such as prevention of sexually transmitted infections among men who have sex with men (MSM). Additionally, many (but not all) transgender and non-binary youth may wish to access gender-affirming healthcare, such as gender-affirming hormones, or discuss future options for this care with their primary care provider [4,5]. Although the number of services is limited, some regions of Aotearoa NZ also provide youth health services through Youth One Stop Shops (YOSSs), [6] community-

based services or student health services in tertiary education settings, some of which are tailored to meet the needs of rainbow youth [7,8].

Rainbow youth globally and in Aotearoa NZ comprise a diverse population who enjoy high levels of involvement and activism in their communities, and who demonstrate resilience in the face of homophobia, biphobia and transphobia [9,10]. However, many also experience significant mental distress and poor physical health outcomes compared to the general youth population [3,11]. This can result from the enduring stigma towards gender and sexuality diversity that can impact their access to and experiences within social and health services, such as compromised trust in providers and avoidance of care [12,13]. The social marginalization of sexuality and gender diversity creates and heightens minority stress for rainbow people, [14,15] leading to compounding physical and mental health impacts [16,17]. Research from Aotearoa NZ shows that rainbow secondary school-aged young people report higher rates of depression, suicidality and self-harm than their cisgender heterosexual peers [3]. Furthermore, the transgender population (including youth and adults) have a rate of mental distress seven times higher than that of the general population in Aotearoa NZ [11].

Several of the inequities that rainbow people experience in both healthcare access and health outcomes, also stem from normative assumptions about gender and sexuality that are implicitly present in healthcare systems [18,19]. Cis/heteronormativity describes the ways in which being cisgender and heterosexual are assumed to be the default or 'normal' in Western cultures, thereby reinforcing the stigma or invisibility of rainbow people [12,20]. In contrast to overt forms of discrimination, cis-heteronormativity tends to manifest as an absence or lack of recognition of gender or sexuality diversity in healthcare structures and interactions. These norms are evident in patient management systems that do not collect data on gender identity or sexual orientation, thereby compromising trans and non-binary people's self-determination [21]. They are also evident in the absence of rainbow-specific content in health professional education that can limit healthcare providers' ability to provide competent care to rainbow communities [22].

Healthcare providers (HCPs) in primary care (e.g., general practitioners, nurses, nurse practitioners) can play a role in determining health outcomes and equity for rainbow populations. Studies in Canada, Australia and Aotearoa NZ involving transgender and non-binary people have shown that people who report feeling respected by and comfortable with their HCP are more likely to report positive mental wellbeing [4,23]. Conversely, people who have had to educate their HCP about rainbow health needs report experiencing higher levels of discrimination and feeling less respected [24] and are more likely to postpone or avoid seeking care [4,25]. The extent to which rainbow people are affirmed in primary care settings often depends on the existing knowledge and attitudes HCPs hold in regard to providing responsive care to both youth and rainbow populations [26,27].

Although initiatives to increase rainbow-specific content in health education are growing, there is a generally low level of knowledge on rainbow communities' health needs among HCPs due to the lack of rainbow-specific content in medical education [28–30]. To date, no qualitative research has been undertaken that centers rainbow youth experiences of primary care in Aotearoa NZ. To address this gap, the present study aimed to explore the experiences and perspectives of rainbow youth when engaging with primary care.

## 2. Materials and Methods

### 2.1. Theoretical Framework

Our study design was underpinned by tenets of critical theory, a broad family of philosophies used in social and youth health research to identify and challenge the norms and power relations that maintain inequities, with the goal of emancipation, empowerment and social justice [31]. Specifically, trans and queer critical theory aims to critique how binary norms around gender and sexuality maintain inequities in settler colonial societies [32,33]. Drawing on these theoretical positions enabled us, as researchers working towards improving health equity for rainbow youth, to understand how participants' ex-

periences of healthcare were enabled or inhibited by the power relations that exist within and beyond healthcare settings. In the context of Aotearoa NZ, it was important for us to understand these relations as products of colonization, and consequently to be explicit about including and reflecting on indigenous knowledge of gender and sexuality, such as these aspects of identity being embraced by Māori pre-colonization [34]. Emphasizing self-determination and principles of youth development [31], we aimed to center the lived experiences of rainbow youth participants in this study, who are experts of their own lives. Our study was further informed by appreciative inquiry [35], a model originally used to understand what is working well within organizations, as well as recognizing issues needing to be addressed. In the context of our research, we took care to understand and communicate participants' experiences, both positive and challenging.

### 2.2. Participants and Recruitment

Two focus groups were held with rainbow young people, as part of a larger project exploring the views and experiences of rainbow youth and primary care staff receiving and providing rainbow-affirming primary care, respectively. Focus group methodology was chosen as it offers unique ways of hearing the experiences of people belonging to marginalized groups and can encourage people to take part when they might not otherwise feel comfortable doing so in an individual interview [36]. A focus group question guide was used for both groups (see Supplementary Material File S1—Interview schedule). Ethical approval was obtained from the University of Otago, Human Ethics Committee (Ref: H21/014, 26 February 2021).

Participants were recruited with assistance from local rainbow support organizations in the Wellington region, through social media and researchers' communication with rainbow support group coordinators. People were eligible to take part in either focus group if they self-identified as rainbow/LGBTQIA+, takatāpui (Māori of diverse sexualities, gender and sex characteristics) or MVPFAFF+ (Pacific peoples of diverse sexualities, gender and sex characteristics); had accessed primary care in Aotearoa NZ; and were aged between 13 and 26. Ethical approval was gained for people aged 16 and over to provide consent to participate without parental consent and, if aged 15 or younger, people had to provide parental consent. In our study, all participants were aged 16 or older, except for one 13-year-old who participated with parental consent.

### 2.3. Data Collection and Analysis

The first focus group was held in-person (July 2021), and the second held via Zoom due to Aotearoa NZ's COVID-19 restrictions at the time (September 2021). Both focus groups were co-facilitated by AK and TG, who are part of rainbow communities and have experience in qualitative interviewing and focus group facilitation with young people. To maintain consistency between groups, AK and TG used the same introductory process and core questions to guide both groups. As expected in focus group methods, the discussions played out differently—perhaps influenced by the nature of existing relationships between some participants and the online and offline formats, which researchers have noted elsewhere impact the quality of focus groups throughout COVID-19 [37]. Participants in Group 1, some of whom were friends, tended to self-moderate their discussion and needed fewer prompts from facilitators, while Group 2, undertaken via Zoom, was guided more by the facilitators; to our knowledge, participants had no existing relationships. The content and emotional tone of the two focus groups differed in some ways with the larger Group 1 collectively sharing more critical perspectives of HCPs overall. Conversely, Group 2 had a more relaxed and positive tone. In both groups, the discussion largely circled around one dominant speaker which may have guided the conversations into a particular, converging narrative. Both focus groups were audio-recorded and transcribed verbatim by a transcriber, who signed a confidentiality agreement form. Each participant was given a $20 gift voucher in acknowledgement of their participation.

The transcripts were analyzed by AK and TG using reflexive thematic analysis, an approach to qualitative data analysis that provides a flexible framework for exploratory studies [26]. We took a largely inductive approach, noting that inductive analysis is inevitably influenced to some extent by researchers' epistemological assumptions [38]. AK initially undertook manual complete coding of both transcripts and sorted both semantic and latent codes into a coding scheme. From this, AK and TG grouped codes into concepts and candidate themes. The themes were identified, refined, reviewed and confirmed by the wider research team. Our research team is a group of predominantly Pākehā (NZ European) female researchers, clinicians and community advocates of various ages from diverse disciplinary backgrounds including primary care, nursing, occupational therapy, youth health, mental health and sociology. Most of the team identify as cisgender and heterosexual, and the two members who led the data collection and analysis are "insider researchers" [39], part of rainbow communities. We were cognizant that each member's interpretation of the data was informed by their prior knowledge, experience and social positioning to the research topic in terms of age, gender and sexuality [40]. For example, when discussing which aspects of participants' accounts were specific to their rainbow identities, the rainbow team members reflected on personal and anecdotal experiences regarding rainbow young people's experiences of healthcare, and the researchers with clinical youth health experience shared knowledge about previous youth research and/or youth healthcare provision in general. Combined, these perspectives informed our discussion of the data and the findings we identified.

## 3. Results

### 3.1. Participants

Eight young people attended Group 1 (110 min in duration) and three young people attended Group 2 (74 min), with one participant attending both groups. The age range of participants was 13–23 years (mean age 18 years). Participants described their gender and sexuality in diverse ways. Table 1 describes the demographic characteristics of focus group participants.

**Table 1.** Characteristics of focus group participants.

| Self-Reported Characteristics [a] | Group 1 (n = 8) In-Person | Group 2 (n = 3) Zoom | Total (n = 11) |
|---|---|---|---|
| | n | n | n |
| Age | | | |
| 13–17 [b] | 5 | 2 | 7 |
| 20–23 | 3 | 1 | 4 |
| Ethnicity (total count) [c,d] | | | |
| European, Pākehā | 7 | 2 | 9 |
| Māori | 2 | 1 | 3 |
| Chinese | 2 | 0 | 2 |
| Gender (total count, verbatim) [c] | | | |
| Non-binary | 2 | 1 | 3 |
| Non-binary trans masc | 1 | 0 | 1 |
| Tāhine, Non-binary | 1 | 0 | 1 |
| MtF trans | 1 | 0 | 1 |
| Male trans | 1 | 0 | 1 |
| Male | 1 | 2 | 3 |
| No idea | 1 | 0 | 1 |
| Sexuality (verbatim) | | | |
| Bisexual | 1 | 0 | 1 |
| Gay | 2 | 0 | 2 |
| Lesbian | 1 | 1 | 2 |
| Not straight | 1 | 0 | 1 |
| Panromantic/asexual | 0 | 1 | 1 |
| Panromantic/sexual | 1 | 0 | 1 |
| Pansexual | 1 | 0 | 1 |
| Queer | 0 | 1 | 1 |
| Unlabelled | 1 | 0 | 1 |

[a]. These are the terms used by the participants to describe their gender and sexuality. [b]. One participant was aged 13 years and participated with parental consent, all others were 16 years or older. [c]. For both ethnicity and gender, some participants indicated more than one ethnicity or gender descriptor so the total count sums to more than 11. [d]. European, Pākehā category includes: NZ European (4), European (1), British (1), Pākehā (2), Irish (1).

Four main themes were identified across focus groups, relating to (i) anticipated and enacted discrimination, (ii) building trust, (iii) confidentiality and (iv) healthcare provider knowledge and competence (see Table 2). Themes and subthemes are discussed further below and illustrated by pertinent quotes from participants. All themes represent patterns across both focus groups.

**Table 2.** Summary of themes and subthemes.

| Themes | Subthemes |
|---|---|
| Anticipated and enacted discrimination | Cis-heteronormativity and lack of understanding |
| | Positive care as "lucky": The anticipation of negative experiences |
| Building trust | Clear communication and explanation build trust |
| | Displaying rainbow-inclusive signs |
| Confidentiality | |
| Healthcare provider knowledge and competence | |

*3.2. Theme One: Anticipated and Enacted Discrimination*

Overall, participants shared both positive and negative experiences of primary care and mixed views towards HCPs. Most reported some experience of being misgendered, stereotyped, or unsupported in primary care settings as rainbow young people. Due to these personal experiences, and hearing about experiences of their rainbow peers, some participants talked about anticipating negative treatment in future interactions with HCPs or described positive experiences as "lucky" or "surprising". Positive experiences with accessing rainbow-friendly primary care included open-minded HCPs who took a person-centered approach to providing care.

3.2.1. Cis-Heteronormativity and Lack of Understanding

A number of participants across both focus groups noted how norms about gender and sexuality were enacted by HCPs. The assumptions that all people are by default heterosexual and/or cisgender led some participants to express that they felt their gender or sexuality was considered not 'normal' in their care provision. Participants identified these norms in interactions (i.e., some HCPs' comments), and in enrolment forms that did not collect data on gender diversity.

> In my old practice, everyone was automatically assuming [ . . . ] you must be straight and so whenever you go to the doctor, it's like he tried to talk to you about things that are quite heteronormative, like if you're a girl and you're gonna have, you know, sex with this guy, you need to do this in order to not get pregnant or to not do that and it's like they don't really take into account that not everyone is straight. (C, Group 2)

As some participants noted, the impacts of these norms may be compounded for young people due to assumptions made about their age and developmental stage. For example, several participants in Group 1 agreed with J when they said, "*lots of doctors and things like that are still in the mindset of you're too young to know that you're gay but you're old enough to know that you're straight.*" (J, Group 1)

Alongside hetero- and cisnormativity, some participants shared that a perceived lack of understanding around gender or sexuality diversity also affected how HCPs responded to their rainbow identity. This included HCP comments about a person's appearance, or assumptions about having multiple sexual partners, that the young people considered inappropriate, or lack of understanding about gender dysphoria.

> I sometimes struggle with the bedside manner of my doctor when talking about things that are very upsetting to me personally but obviously to a cis[gender]

person, they don't understand why they're upsetting to me [...] I'm like, maybe you don't understand how dysphoria works but it's actually quite painful like and I would prefer you not to, you know, joke around with me about it when you don't really know what I'm going through. (E, Group 1)

A few trans and non-binary participants from each group spoke about how their HCP's lack of understanding about trans people's health needs had caused them to doubt the validity of their gender identity or to delay medical transition. Participant C in Group 2, for example, shared that when they approached their doctor about accessing gender-affirming healthcare, "*[my doctor's] response was, kind of, 'you are what you are at birth and there's no getting around that'*".

### 3.2.2. Positive Care as "Lucky": The Anticipation of Negative Experiences

The positive experiences that some participants reported were characterized by HCPs being accepting and open-minded about the young person's gender or sexuality (if they had disclosed this) and following the young person's health requests such as accessing gender-affirming healthcare. These experiences tended to happen in youth health services. For some participants in both groups, the expectation of negative experiences meant that these positive interactions, when they happened, were seen as unexpected. This is reflected in participants' descriptions of positive experiences as "surprising", "lucky" or "cheating".

Participants' anticipation of negative treatment was often attributed to their own prior healthcare encounters or hearing about their rainbow peers' experiences. This fear of negative treatment created a heightened awareness of potential discrimination within primary care settings, even when the enacted care was otherwise positive or affirming:

G: I think when you have a bad experience, it makes you hypervigilant to the bad stuff happening again. Like even though my current GP, she's amazing [ . . . ] no matter how long I'm with this new GP, there's always the fear that it's gonna happen again.

Several: Yeah.

L: Yeah, cause you always remember the bad experiences. (Group 1)

As G's and L's comments illustrate, the impacts of previous negative experiences for rainbow young people can inform how they experience positive care provision.

### 3.3. Theme Two: Building Trust

Participants in both groups valued trustful and consistent relationships with their HCPs, yet previous negative experiences often created an overall sense of distrust toward HCPs. Some expressed uncertainty around how HCPs would react to their gender or sexuality if they were to disclose it, and the fear of being asked invasive or irrelevant questions put some participants off discussing their rainbow identity in the first place.

### 3.3.1. Clear Communication and Explanation Build Trust

When participants spoke of positive experiences of trust in primary care, they associated this with clear communication and explanations. This sort of communication made them feel comfortable and safe in contrast to poor communication, which resulted in distrust and uncertainty making participants feel awkward or uncomfortable. The following was an example of clear HCP communication:

When I went in for like a random thing to the doctor, it was a nurse that I hadn't had before and she like sat me down, talked about all these different ways we could go about doing things and it made me feel really safe and I knew that if she was the one that I booked in to talk to about hormone therapy or whatever, that I would feel really comfortable talking to her about that kind of stuff. (B, Group 2)

Some participants described consultations that were uncomfortable or mistrustful, particularly when HCPs asked them what they perceived to be irrelevant or invasive

questions. Participants expressed that they sometimes did not know how these questions related to sexuality or gender and wanted to have this explained before answering.

> My doctor makes me list off every single side effect of T [testosterone] and I've been on T for like a really long time [ . . . ] it's quite bizarre that she makes me list it off every single time when it's not relating to what I'm talking about at all. (E, Group 1)

There was a shared understanding that disclosing or discussing gender, sexuality, name or pronouns in a consultation could open them up to uncomfortable, invasive, or irrelevant questions:

> F: Is it worth it to crack open the gender egg in the doctor's office . . .
>
> K: Yeah, because then you'll be there for another half an hour talking about something that's not relevant to your back pain, it's talking about what you have down there. (Group 1)

A few participants spoke about their decision to disclose their identity as pragmatic, often depending on the existing relationship and level of comfort they had with their HCP. *"I've actively decided not to [disclose my gender], cause it's just like, it's so much"*. (L, Group 1)

### 3.3.2. Displaying Rainbow-Inclusive Signs

Health services are often encouraged to signal rainbow-friendliness by displaying visual signs such as rainbow flags in waiting areas. This approach garnered mixed feelings across both groups; some participants saw them as a welcome sign of support, while others viewed them as potential "false advertising." Most participants in Group 1 took issue with the "false sense of security" created by rainbow visibility markers when this support was not backed up in practice, with some participants saying they would be less trusting of a clinic which openly displays being rainbow-friendly:

> E: Something that I find really frustrating with misgendering is that at my health clinic, there are signs everywhere talking about, 'tell us your pronouns' [ . . . ] and then just, my top surgery referral was, I was misgendered [ . . . ] I like told my GP and she was just like, 'oh I'm sorry that happened'. I'm like, 'you wrote the letter [that misgendered me], it didn't [just] 'happen', apologize for doing it.'
>
> Facilitator: Like performative.
>
> E: Yeah, it's very performative, very like, 'here's your rainbow tick and we don't actually have to do any like work underneath or the performance' and I find that just like frustrating. (Group 1)

One participant from Group 2, however, thought that it was comforting to see rainbow visibility signs at a clinic and took it as a sign of support and acceptance. Some participants in Group 1 agreed that visibility can be positive, but only when backed up in practice, again affirming that trust and authenticity is essential when communicating rainbow-friendliness.

### 3.4. Theme Three: Confidentiality

There was concern expressed by some participants that HCPs would disclose details about their sexuality or gender to the participant's parents or other staff members. Confidentiality and privacy in primary care settings were expressed as important by many participants. In particular, the younger participants who were dependent on their parents or caregivers when accessing primary care (including paying for appointments or transport), were worried information may be shared with their parents. The nature of relationships between participants and their parents or caregivers often impacted how open participants could be about their rainbow identity or specific health needs. For example, participants with more supportive parents expressed they were able to access primary care more easily and worry less about staff breaching their privacy:

> My mum is like a hard-core feminist, she like goes out of her way to research all of this stuff [ . . . ] and so I've kind of like had very little issues with a lot of medical stuff. (L, Group 1)

On the other hand, participants who described less supportive parents expressed hesitancy to talk about their sexuality or gender with their HCP, due to concerns about the possibility of their parents finding out about their rainbow identity. A few participants in Group 1 expressed fear of parents' negative reactions to them disclosing their gender or sexuality, such as threats of being 'kicked out', which they implied impacted their decisions not to disclose their identities in healthcare settings. This was seen to impact young people's sense of agency over their health decisions. A few participants from each group noted the tensions between the young person's rights to healthcare yet needing caregivers' consent if they were under 16 years old.

### 3.5. Theme Four: HCP Knowledge and Competence

Participants shared that they wanted HCPs to have the healthcare knowledge to meet rainbow young people's needs. Some, who accessed care through youth-friendly care providers, discussed experiences of receiving competent care. Others expressed frustration that when their HCP demonstrated a lack of understanding of rainbow communities, they felt they had to explain gender and sexuality diversity and associated healthcare needs. This expectation to educate was viewed as exhausting and often ineffective; as G (Group 1) noted, *"I did have to educate my old GP but then he just sort of ignored it and never brought it up again."*

Several participants in Group 1 shared that they had little choice but to present their HCPs with their own research, particularly with information about gender-affirming healthcare. For some, it undermined the expectation that HCPs should be equipped to provide appropriate healthcare.

> I feel like you should not be coming to them with the research at all. You should go, 'hey I want to go on testosterone', they should research and 'ok so this is what I know, tell me what you know, we can figure something out. (K, Group 1)

Participants in both focus groups thought that HCPs needed to be educated on providing rainbow-friendly healthcare, both to enable them to provide accurate medical information, and to unlearn the stigma of and normalize rainbow communities. This education was perceived to help improve primary care experiences for rainbow youth.

> [HCPs] need to have more education that there is a whole bunch of people out there that are different from what society deems as like, I guess, the norm. (C, Group 2)

### 3.6. Application of Findings to Primary Care Practice

Table 3 presents recommendations that participants made during the discussions, for things they thought would help HCPs provide effective and responsive care for rainbow young people.

**Table 3.** Participants' recommendations for healthcare providers (HCPs).

| Recommendation | Illustrative Quotes [a] |
| --- | --- |
| Let conversations around our gender and sexuality be led by us. *Listen to and believe young people; take their lead about their health needs and goals.* | - I think that when it's appropriate to bring [gender and sexuality] up, it can be brought up in a non-awkward and [non] invasive way. (H) <br> - If they just sat down and listened to you. (K) <br> - Letting the person going to the doctors take the lead. (C) |

**Table 3.** *Cont.*

| Recommendation | Illustrative Quotes [a] |
|---|---|
| Learn some basics about rainbow health so that we don't feel we need to educate you. *You don't need to know everything but take time to learn the basics. If you are unsure, be honest and say that you will find out, or use a partnership model ("let's find out together").* | - Just learn some of the ABCs of it and stuff like that and learn that it's not all about sex, you know. (H) <br> - They need to have more education that there is a whole bunch of people out there that are different from what society deems as like I guess the norm and they need to be open and be expecting that because we're not just gonna like disappear. (C) <br> - I feel like you should not be coming to them with the research at all. You should go, 'hey I want to go on testosterone', they should research and 'ok so this is what I know, tell me what you know, we can figure something out'. (K) |
| Respect our gender or sexuality. *Try to avoid heteronormative and cisnormative language, making assumptions and misgendering people (e.g., if you don't know someone's gender/pronouns use neutral terms like they/them).* | - Not being invasive. (K) <br> - Not being misgendered on a constant basis. (E) <br> - Don't just automatically assume that someone's one thing because that's how everyone else in society is. (C) |
| Provide services that are responsive to our needs. *Undertake a co-design or consultation process with young people when services are being developed.* | - It could even be on like the doctor's (web)page or something, be like they have like a youth section and then just sort of like you know, you can have all of those frequently asked questions and things like that. (H) <br> - We need more [youth health services] please. (K) |
| Take the time to build trust with us. *Making time to undertake whakawhanaungatanga or introductions is critical to building trust with young people. It takes time and practice to do this well. Trust is foundational to facilitating disclosure* [41]. | - I think another thing that could build trust is people who are doctors and things like that, when they walk into the office in the morning leave everything, like religion, anything like that outside. (J) <br> - It needs to have the same kind of like professional boundary where you don't make the kind of like sexual jokes over your comments or that kind of thing, but it is still friendly and it is still open. (F) |
| Maintain our right to privacy. *Communicate what confidentiality means at several points during the consultation and how this applies to sharing/not sharing with parents. Clarify the situations in which confidentiality can be broken and when disclosure has to occur.* | - I think something as well especially if it's like a family GP, so you know, all the whole family seeing it. Don't cross over information and stuff. (H) <br> - If they give you a list and you go, 'look this is what I legally have to tell you but if you tell me not to say anything else, then I cannot say it', that's how it should go. (K) |
| Check in if it's okay to ask about something before asking and explain why you are asking. *E.g. Explain that all young people are offered opportunistic STI screening (due to high rates of asymptomatic infection).* | - Actually give a warning of the sort of things that might actually happen in that specific appointment or whatever and ask instead of assuming. Ask if there's preferences or something. (D) <br> - Don't just like throw it onto the person without explanation, I guess. (C) |

[a]. Letters in brackets refer to individual participant identifiers.

## 4. Discussion

Participants in our study reported mixed experiences of primary care. Positive experiences often involved youth being listened to and not judged by HCPs, accessing care through youth health services, and parental support. However, participants expressed evident caution in primary care settings, particularly around disclosing their rainbow identity, due to the anticipated judgement or lack of understanding from HCPs about rainbow young people's identities and health needs. This resulted in positive care experiences being perceived as the exception, rather than the rule. Participants expressed that they valued

clear communication, trust that their confidentiality would be respected, and staff who were educated on gender and sexuality diversity. When they were met with HCPs who explained why they were asking certain questions, and an assurance of confidentiality, participants were more likely to experience this as a trusting relationship and feel comfortable to disclose their gender identity or sexuality to their HCP. They shared that they deserved to be treated "normally", implying that they felt they were treated differently, or unfairly, as rainbow people in healthcare settings.

The negative experiences that we identified are consistent with a US mixed methods study [42,43] that also identified HCPs' inappropriate comments, and lack of clear communication and confidentiality, as undermining rainbow young people's primary care experiences. Several of the barriers to good quality and affirming care that we identified among rainbow youth are similar to those that cisgender and heterosexual youth also report in research elsewhere, such as confidentiality, comfort with HCPs, relationships with caregivers, and youth-friendliness of service [44–46]. However, our findings highlight that rainbow youth experience these barriers with an additional layer of caution and minority stress due to the pervasiveness of cisnormativity and heteronormativity, and consequent discrimination, in both healthcare settings and in wider social domains such as family and school environments. Experiences of compromised trust that result from these norms have been identified among younger men who have sex with men [47], non-binary service users [48] and transgender youth who are misgendered in healthcare settings [42,43]. Experiences and the anticipation of negative experiences have been shown elsewhere to result in avoiding care or not disclosing one's gender or sexuality in care settings [11,47], thus perpetuating the disparities in health outcomes among rainbow populations. Building on this evidence, our findings suggest that the cis-heteronormativity that rainbow young people experience in their lives, sometimes in previous healthcare experiences, adversely influence their expectations of primary care and thus how comfortable they are entering these settings.

From a health equity perspective, these findings have important implications for primary care provision in understanding the unique barriers rainbow youth experience to safe and affirming care. It is imperative that HCPs receive education about the importance of not making assumptions about people's gender identity, sexuality, name and pronouns, as well as more general upskilling in health topics relevant to rainbow populations. Making simple changes to create safer and more welcoming environments for rainbow service users, such as using gender-inclusive language, correct pronouns and names [49] and providing rainbow-friendly visual cues in the service, have been shown to facilitate disclosure [50,51] and increase patients' positive perceptions of and access to care [48]. Our study shows it is essential, however, to back up any visual cues with an inclusive and accepting practice, to maintain trust and ensure these cues are not seen as a token gesture.

When the experience was positive, participants felt listened to, not judged, and respected by primary care staff. Existing studies on transgender and non-binary people's experiences of primary care have also found that positive experiences tend to be characterized by respectful, relational and non-judgmental care; [43] providers who are approachable, friendly and relaxed; [52] and youth feeling that their needs and experiences are heard [53]. Being clear about confidentiality is important when creating positive experiences for rainbow youth accessing primary care [42,54]. Considering that positive healthcare experiences are associated with better mental health outcomes for rainbow populations [4], working towards achieving consistency of affirmative care would likely play an important role in improving health outcomes for this population. However, our study highlights that positive experiences are often perceived by rainbow youth as a matter of "luck" or exceptional circumstance rather than a baseline expectation of care, suggesting that more work is required to improve the consistency of positive experiences, to minimize the hypervigilance rainbow youth often face when accessing care.

*Strengths and Limitations*

This study is one of the first to report on primary care experiences of rainbow youth in an urban setting in Aotearoa NZ. Our participants were diverse in age, ethnicity, gender and sexuality, and were generous in sharing their perspectives and experiences with primary care. While research of this nature has been conducted elsewhere, country-specific research is important to acknowledge unique cultural characteristics, local context and provision of primary care in Aotearoa NZ. These study findings, and particularly the recommendations generated by participants, can be used to help guide HCPs in their provision of more equitable healthcare for this underserved population. Use of focus group methodology and facilitators who are members of rainbow communities were strengths of this work, by providing a sense of shared experience for participants and facilitators, and in creating an environment where young people felt comfortable to talk openly about their experiences with the support from other focus group participants.

Limitations of our study include the relatively small sample that was primarily drawn from one region - an urban setting with a reputation for being socially progressive. For this reason, we acknowledge that the perspectives shared by participants in this study will not be reflective of all rainbow young people in Aotearoa NZ. Further, we were unable to comment on differences in healthcare experiences based on participants' different identities (such as non-binary and binary trans participants) and acknowledge that further in-depth research is warranted on this topic. Our study included only a small number of Māori participants so our ability to contribute to discussion on aspects such as cultural responsiveness to takatāpui and Māori rainbow young people in primary care is limited. While many takatāpui and Māori rainbow young people are loved and supported by whānau, others can experience compounding effects of racism and homo/bi/transphobia [34]. Further research focusing specifically on takatāpui and Māori rainbow young people's experience of primary care is needed to inform more culturally responsive service provision.

## 5. Conclusions

Rainbow young people believe HCPs can contribute to positive healthcare experiences when they are accepting and respectful, avoid making assumptions, are transparent about confidentiality and use clear explanations to build trusting relationships. Improving healthcare equity for this population requires education on rainbow communities' unique health needs and experiences, and resources to support HCPs to challenge harmful stereotypes about gender and sexuality evident within health systems and broader social contexts that can shape rainbow young people's healthcare interactions.

**Supplementary Materials:** The following supporting information can be downloaded at: https://www.mdpi.com/article/10.3390/youth2040049/s1, Supplementary Materials File S1: Interview schedule.

**Author Contributions:** Conceptualization, R.C., A.K., T.G., S.B.R., S.J.M., S.M.G. and E.M.M.; methodology, R.C., A.K., T.G., S.B.R., S.J.M., S.M.G. and E.M.M.; formal analysis, A.K. and T.G.; investigation, A.K. and T.G.; data curation, A.K., T.G. and S.J.M.; writing—original draft preparation, A.K.; writing—review and editing, A.K., R.C., T.G., S.B.R., S.J.M., S.M.G. and E.M.M.; funding acquisition, R.C., A.K., T.G., S.B.R., S.J.M., S.M.G. and E.M.M. All authors have read and agreed to the published version of the manuscript.

**Funding:** This research was funded by a University of Otago Research Grant (2021) and a Lotteries Health Research Grant (2021–2022).

**Institutional Review Board Statement:** Ethical approval was obtained from the University of Otago, Human Ethics Committee (Ref: H21/014, 26 February 2021).

**Informed Consent Statement:** Informed consent was obtained from all participants involved in the study.

**Data Availability Statement:** Raw data are not publicly available as study participants were not asked for permission, and nor was ethical approval granted for data to be shared publicly.

**Acknowledgments:** We thank the young people who participated in the two focus groups for sharing their time, views and experiences.

**Conflicts of Interest:** The authors declare no conflict of interest. The funding bodies had no role in the design of the study; in the collection, analyses, or interpretation of data; in the writing of the manuscript, or in the decision to publish the results.

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
