# Peer review of "“We Just Want to Be Treated Normally and to Have That Healthcare That Comes along with It”: Rainbow Young People’s Experiences of Primary Care in Aotearoa New Zealand"

_2673-995X, doi:10.3390/youth2040049_

Round 1

Reviewer 1 Report

Thank you so much for inviting me to review this article, which is an important piece of work in LGBT studies. I have few comments for improving the final version, but you do not necessarily need to agree with me:

1) Regarding the ethical issues of interviewing adolescents under the age of 18, it will be nice if the authors will write briefly about this to the readers for an understanding. Are parental consents needed in your country for those aged 16-18? what is the age of consent for research? Are the parental consent waived for those aged 16-18 if the age of consent is beyond 16 years old?

2) In the result session, the authors demonstrated the themes of anticipated and enacted discrimination, cisnormativity , it will be nice if these terms or related studies will be discussed in literature review, and connected to the discussion section.

3) What is the theoretical framework or areas of knowledge informing your study? It will be nice to highlight to your readers in few lines briefly in your introduction or literature session.

Thank you very much for giving me this opportunity to read your research article, very valuable study. 

Author Response

We thank our reviewers for taking the time to read and review our submission, and for their provision of helpful comments and suggestions. We have addressed each point below, and made corresponding changes where relevant to our manuscript.

Reviewer 2 Report

This is a very interesting paper looking at young people’s experiences of primary care in Aotearoa New Zealand.

While I concur that substantially more research attention is needed on and with LGBT+ individuals, I have some concerns about this manuscript in its present state, which dampen my enthusiasm for the research overall.

Below, I outline my comments and concerns for the authors.

INTRODUCTION

Generally speaking, I didn't feel as though enough research had been cited in the introduction. For example, I was surprised that West and Zimmerman's (1984) "Doing Gender" wasn't cited, or Morgenroth & Ryan's (2019) "The Effects of Gender Trouble."

Since the study involved people belonging to different identities, I think It could be useful to specify differences between transgender, non-binary and genderqueer people, also citing:

Tatum, A. K., Catalpa, J., Bradford, N. J., Kovic, A., & Berg, D. R. (2020). Examining identity development and transition differences among binary transgender and genderqueer nonbinary (GQNB) individuals. Psychology of Sexual Orientation and Gender Diversity, 7(4), 379-385. https://doi.org/10.1037/sgd0000377

Whyte, S., Brooks, R. B., & Torgler, B. (2018). Man, woman, "other": Factors associated with nonbinary gender identification. Archives of Sexual Behavior, 47, 2397-2406. https://doi.org/10.1007/s10508-018-1307-3

METHOD

1) What was the focus group setting like? How were the characteristics of the setting replicated in the online focus group?

I suggest reading and citing the following works, which may provide interesting insights to highlight the differences and similarities between online and offline research involving sexual and gender minorities:

Monaco, S. (2022). Gender and sexual minority research in the digital society. In Handbook of research on advanced research methodologies for a digital society (pp. 885-897). IGI Global.

Monaco, S. (2022). Researching Parenting in Pandemic Times: Tips and Traps from an Italian Experience. Italian Sociological Review, 12. DOI: http://dx.doi.org/10.13136/isr.v12i7S.585 

2) The authors assert that they have engaged with thematic analysis in the paper, yet the authors make no mention of their reflexivity process or critical consideration of how their own identities may have influenced the research process, which is the core of reflexive thematic analysis. I often recommend the following readings when I discuss  thematic analysis:

Braun, V., & Clarke, V. (2021). One size fits all? What counts as quality practice in (reflexive) thematic analysis? Qualitative Research in Psychology, 18(3), 328-352. https://doi.org/10.1080/14780887.2020.1769238

For practicing explicit positionality, I encourage you to consider employing Jacobson & Mustafa's tool:

Jacobson, D., & Mustafa, N. (2019). Social identity map: A reflexivity tool for practicing explicit positionality in critical qualitative research. International Journal of Qualitative Methods, 18. https://doi.org/10.1177/1609406919870075

3) The methodological and epistemological assumptions underlying this specific thematic analysis are not made clear in the paper. In Braun and Clarke's (2006), the authors explicitly state that the authors must articulate their approach to the analysis: inductive or theoretical? Latent or semantic? Constructivist or realist? While thematic analysis is beneficial for its flexibility, the assumptions underlying the analysis must be made clear.

RESULTS

1) When presenting quotes to support the themes, I broadly prefer to see them integrated into a narrative, complete with thick descriptions of the participants and the contexts under which the quotes were delivered.

2) I would like to see a table with codes, themes and subthemes clearly laid out for the reader.

Finally, there are some grammatical errors that limit its readability. I encourage the authors to read through the paper with an eye for cohesiveness and flow.

Round 2

Reviewer 2 Report

I very much appreciated the authors' effort in improving their paper. They applied most of the directions I had given them. 

Although they could have further strengthened their paper, I believe that the revised version can be considered solid enough to be published in the journal.